# Energy and Nutrient Intakes of Public Health Concern by Rural and Urban Ghanaian Mothers Assessed by Weighed Food Compared to Recommended Intakes

**DOI:** 10.3390/nu17152567

**Published:** 2025-08-07

**Authors:** Prince K. Osei, Megan A. McCrory, Matilda Steiner-Asiedu, Edward Sazonov, Mingui Sun, Wenyan Jia, Tom Baranowski, Gary Frost, Benny Lo, Christabel A. Domfe, Alex K. Anderson

**Affiliations:** 1Department of Nutritional Sciences, College of Family and Consumer Sciences, University of Georgia, Athens, GA 30602, USA; fianko@uga.edu; 2Department of Health Sciences, Sargent College of Health & Rehabilitation Sciences, Boston University, Boston, MA 02215, USA; mamccr@bu.edu; 3Department of Nutrition and Food Science, College of Basic and Applied Science, University of Ghana, Legon, Accra P.O. Box LG 25, Ghana; tillysteiner@gmail.com; 4Department of Electrical and Computer Engineering, University of Alabama, Tuscaloosa, AL 35487, USA; esazonov@eng.ua.edu; 5Department of Neurological Surgery, University of Pittsburgh, Pittsburgh, PA 15213, USA; drsun@pitt.edu; 6Department of Electrical and Computer Engineering, University of Pittsburgh, Pittsburgh, PA 15261, USA; jiawenyan@gmail.com; 7USDA/ARS Children’s Nutrition Research Center, Department of Pediatrics, Baylor College of Medicine, Houston, TX 77030, USA; tbaranow@bcm.edu; 8Section for Nutrition Research, Department of Metabolism, Digestion and Reproduction, Imperial College London, London W12 0NN, UK; g.frost@imperial.ac.uk; 9Hamlyn Centre, Department of Surgery and Cancer, Imperial College London, London SW7 2AZ, UK; benny.lo@imperial.ac.uk; 10Department of Nutrition, College of Agricultural and Environmental Sciences, University of California, Davis, CA 95616-5270, USA; cadomfe@ucdavis.edu

**Keywords:** mothers, recommended nutrient intake, dietary assessment, weighed food records, Ghana

## Abstract

**Background/Objectives:** Previous studies assessing dietary intake have used self-report methods, prone to misreporting. Using researcher-conducted weighed food records, we assessed rural and urban mothers’ energy and nutrient intakes of concern and compared them to recommended nutrient intakes (RNIs). **Methods:** This cross-sectional study was conducted in rural (Asaase Kokoo) and urban (University of Ghana Staff Village) communities. Dietary data were collected from fifty-four mothers (26 rural, 28 urban) on 2 weekdays and 1 weekend day, analyzed with software, and programmed with West African, FNDDS, Kenyan, Ugandan, and USDA food composition databases. **Results:** Mean (SD) ages (years) were 35.8 (11.6) and 44.4 (7.6), and mean energy intakes (kcal) were 2026 (461) and 1669 (385) for rural and urban mothers, respectively. Mean percentage contributions of macronutrients to energy intake were within recommended ranges for rural and urban mothers. All participants met or exceeded vitamin A RNI, irrespective of location. While all rural mothers met or exceeded iron RNI, some urban mothers (14.3%) did not. Few rural (7.7%) and urban mothers (10.7%) did not meet zinc RNI. About half of rural (46.2%) and urban mothers (53.6%) did not meet folate RNI. Most rural (96.1%) and urban mothers (92.8%) met or exceeded fiber RNI. **Conclusions:** Overall, rural mothers had higher energy and nutrient intakes than urban mothers. While most met RNIs, there were some micronutrient inadequacies, particularly folate, where almost half of rural and urban mothers consumed below RNI. Our findings indicate the need for tailored interventions to prevent nutrient deficiencies or excesses in Ghanaian mothers.

## 1. Introduction

Malnutrition exists in different forms, including overnutrition (overweight and obesity) and undernutrition (underweight, stunting, wasting, and micronutrient deficiencies) [1]. Over the past decade, the prevalence of overweight and obesity has increased significantly among women in low- and middle-income countries (LMICs) [2]. Micronutrient deficiencies thrive in many countries across the globe, with LMICs bearing the greatest burden because of factors such as inadequate food and nutrient intake, low dietary diversity, food insecurity, and sociocultural and geographical factors [3,4,5]. Likewise, the prevalence of overweight and obesity is ascending in Ghana, while deficiencies in vitamin A, folate, iron, and zinc remain significant public health concerns [6,7]. Among Ghanaian women, the prevalence of overweight and obesity was previously reported to be higher in urban areas (45.8%) than in rural areas (25.3%) [8,9]. Higher prevalence of overweight and obesity, alongside micronutrient deficiencies, was reported among some Ghanaian women [10,11]. To address malnutrition and its associated health implications, the government of Ghana implemented various intervention strategies, including weight management programs, salt iodization, micronutrient supplementation, wheat flour fortification, and the use of micronutrient powder sprinkles [12,13,14]. However, despite these efforts, micronutrient deficiencies persist, and the prevalence of overweight and obesity continues to increase, particularly in urban populations and among women [6,7].

There is currently a dearth of information on the dietary intake of mothers in rural and urban communities of Ghana. Few studies that have assessed dietary intake in the Ghanaian population have relied on self-report methods, such as the 24-h dietary recall [15,16] and food frequency questionnaire [17,18]. However, findings from these self-report dietary assessment methods have been conflicting. Compared to weighed food records, while one study reported that the 24-h dietary recall underestimated energy and nutrient intakes [19], another study observed that the 24-h dietary recall overestimated intakes [20]. Additionally, a study found that the food frequency questionnaire overestimated nutrient intake compared to the weighed food records [21]. These inconsistencies indicate that self-report methods are prone to misreporting, highlighting the need for more accurate dietary assessment tools. To the best of our knowledge in the literature, no study has used researcher-conducted weighed food records (rWFRs), considered the gold standard for dietary assessment, to assess dietary intake in rural and urban mothers at the household level in Ghana. Therefore, we assessed the energy and selected nutrient intakes (protein, fat, carbohydrate, fiber, vitamin A, folate, iron, and zinc) of rural and urban mothers using rWFRs and compared them to the World Health Organization (WHO)’s recommended nutrient intakes (RNIs) [22,23]. The selected micronutrients were chosen due to their significant public health concerns in Ghana [24]. We also compared the percentage contributions of protein, fat, and carbohydrate to the total daily energy intake against the recommended ranges [22].

We hypothesized that the energy intake of both rural and urban mothers would exceed the RNI, with higher energy intake expected among urban mothers compared to their rural counterparts. This hypothesis was based on findings from previous studies reporting that food environments in urban areas of Ghana are generally more saturated with processed and energy-dense foods than those in rural areas [25,26]. We expected that the percentage contributions of protein, fat, and carbohydrate to daily energy intake would fall within the recommended ranges among rural and urban mothers. Given the ongoing public health concerns regarding deficiencies in vitamin A, folate, iron, zinc, and fiber in Ghana [24], we anticipated that the daily intakes of these nutrients would fall below the RNIs in both rural and urban mothers. This study was part of a larger research project to validate passive innovative technologies for dietary assessment in LMICs [27].

## 2. Materials and Methods

### 2.1. Study Design, Eligibility Criteria, and Sample Size

This cross-sectional study was conducted in rural and urban communities of Ghana, targeting mothers in selected households. For the rural community, Asaase Kokoo in the Akuapem North Municipal Assembly in the Eastern Region was selected, while the University of Ghana Staff Village in the Greater Accra Region was the urban community. As part of the larger research project, per the study protocol, eligibility included a household having a mother, father, and an index child (adolescent and/or child under 5 years of age) residing in the same household and consuming most of their meals at home. The full methodology and eligibility criteria are described elsewhere [27]. Purposive sampling was used to select 60 households (30 in rural and 30 in urban), with the assistance of community leaders in identifying eligible households. The main study protocol was approved by the Human Subjects Institutional Review Board of the University of Georgia (STUDY00006121) and the Institutional Review Board Committee of the Noguchi Memorial Institute for Medical Research at the University of Ghana (#-046/18-19). Household members (including mothers) who showed interest in participating in this study provided consent by either signing or thumb-printing informed consent forms.

### 2.2. Data Collection

Data collection was conducted by field staff in households. All field staff were multi-lingual (spoke English and at least one other local language) to facilitate effective communication with participants from diverse ethnic backgrounds in rural and urban communities. Field staff also had at least a bachelor’s degree in nutritional sciences and were trained in standardized protocols and interviewing techniques to ensure accurate data collection. In rural households, data collection began in November 2020 and continued until the end of February 2021, with a break for the Christmas holidays. In urban households, data collection began in March 2021 and ended in May 2021, with a break for the Easter holidays.

On a typical day of data collection, two field staff arrived at the household early in the morning. Upon arrival, the field staff verified that no food or beverages had been consumed. They remained nearby so as not to interfere with the normal daily activities of the household but responded to calls from participants. During food preparation and consumption, the field staff were invited into the household to weigh all raw food ingredients (recipes) that were used to prepare meals, snacks, and portions to be consumed. Foods and beverages consumed by the mother alone, as well as leftovers, and those shared with other household members, were weighed and recorded on record sheets. The identities of individuals who shared foods and beverages with the mother in the household during meal-sharing and eating events were also recorded. Foods and beverages purchased from food vendors outside of the home and consumed at home by the mother were weighed and recorded. The dietary intake assessment focused solely on foods and beverages consumed by participants, including fortified foods, traditional foods, processed foods, sugar-sweetened beverages, and others. Dietary supplement intake was not assessed.

### 2.3. Socio-Demographic Data and Anthropometric Measurement

In addition to dietary intake assessment in the household, the mother participated in a brief interview to complete a survey on the household’s socio-demographic characteristics. The mother’s weight and height were measured using a Seca 769 digital column weighing scale equipped with a stadiometer (Hammer Steindamm, Hamburg, Germany). Height was measured in centimeters, while weight was measured in kilograms, following standard protocol (participants wearing full clothing with no heavy accessories on their bodies and standing upright without shoes on the scale for height and weight measurements the morning of a study day). The measured height and weight were used to calculate body mass index (BMI) (kg/m^2^). BMI values were categorized according to WHO’s weight classification criteria for adults [28]: <18.5 kg/m^2^ as underweight, 18.5–24.9 kg/m^2^ as normal weight, 25.0–29.9 kg/m^2^ as overweight, and ≥30.0 kg/m^2^ as obese. After dinner, when all food preparation and consumption activities were completed for the day, the field staff departed from the home. All data collected were reviewed independently by the project coordinator, who was not directly involved in the data collection process at the household level. Finally, the data were uploaded to the database (password-protected cloud space) of the larger research project.

### 2.4. Nutritional Analysis

Foods and beverages consumed by the mother alone and those shared with other household members were recorded separately and analyzed by one analyst trained in portion size estimation and database item selection using custom AIM annotation software (version 4.3) [29]. The software was programmed with the West African [30], Kenyan [31], Ugandan [32], Food and Nutrient Database for Dietary Studies (FNDDS) [33], and the United States Department of Agriculture (USDA) [34] food and nutrient composition databases. The software provided simultaneous access to all five food and nutrient composition databases, but a prioritization was set to use them in the order stated above.

Dietary data were analyzed in two phases. In the first phase, foods and beverages consumed solely by the mother (i.e., not shared) on a given day were analyzed. In the second phase, foods and beverages that the mother shared with other household members on that same day were analyzed [35]. The analyzed data from both phases for each mother were then combined to represent her total daily intake. A recipe calculation (informed by the ingredients used for meal or beverage preparation) approach was used to analyze foods and beverages that were not directly present in the food composition databases programmed into the software. With this approach, the ingredients in recipe preparation served as a guide in estimating the proportions of ingredients (all ingredients or suitable substitutes were present in the food and nutrient databases used), and the corresponding quantity of dish consumed by the mother. For analysis of foods and beverages shared with other household members, standardized ratios (1:1 for a mother and father sharing a meal, 3:2 for a mother and adolescent sharing a meal, and 3:1 for a mother and child under five years sharing a meal) were assigned to the mother and each household member who shared in the eating of the specific meal or beverage to estimate the amount of food or beverage consumed by each person and, subsequently, the portion size consumed by the mother [35]. These ratios were informed by the judgment and lived experiences of Ghanaian investigators, providing cultural context that reflected typical household food-sharing and eating practices, including age- and gender-specific eating patterns and relative portion sizes.

In the analysis, once each food or beverage item and the corresponding portion consumed by the mother were estimated and entered into the AIM annotation software (version 4.3), it calculated the energy and nutrient content for each item, by eating occasion (breakfast, lunch, dinner, and snack) and for the entire day. The same procedure was applied to analyze the dietary data collected from each mother on two weekdays and one weekend day. Mean energy and nutrient intakes were calculated by averaging the intake values across the three days. The mean values of the outcome variables were then imported into a statistical software for analysis.

### 2.5. Calculations and Statistical Analysis

Statistical analysis was conducted using IBM SPSS Software 29.0 (Armonk, NY, USA). Among the 60 mothers who participated in this study, dietary intake data were incomplete or missing for 4 mothers in rural households and 2 mothers in urban households. Therefore, the data presented in this paper are for 26 rural mothers and 28 urban mothers. Descriptive statistics were calculated for socio-demographic and anthropometric data. The ages and BMI of rural and urban mothers and a moderate physical activity level (based on the Ghana Health Service report [36] on the normal activity level of adult Ghanaian women) were used to determine the RNIs. Group mean daily energy, protein, fat, carbohydrate, fiber, vitamin A, folate, iron, and zinc intakes, as well as the percentage contributions of macronutrients, were compared to the RNIs as outlined in the handbooks of the WHO’s recommended energy and nutrient intakes [22,23]. The mean daily energy and nutrient intakes of rural versus urban mothers were compared using independent sample t-tests, while one-sample t-tests were used to compare intakes to RNIs.

The z-scores for mean daily energy, fiber, vitamin A, iron, folate, and zinc intakes were computed relative to RNIs and were used to determine the percentages of rural and urban mothers who fell below, met, or exceeded the RNIs. They were calculated using the formula: Z-score = (x − RNI)/SD, where x represents each mother’s observed mean intake for a given nutrient, RNI is the recommended intake level for that nutrient, and SD is the standard deviation of intake for that nutrient computed from the study sample [37]. This approach allowed us to assess how far each mother’s intake deviated from the RNI. Using a probability-based classification, mothers with mean z-scores between −1 and +1 were designated as having met the recommended intakes, mean z-scores above +1 indicated intake exceeded the recommendation, while below −1 indicated intake that fell below the recommendation [37,38]. This standardized approach to evaluating nutrient adequacy enabled us to estimate the proportions of rural and urban mothers whose intakes fell below, met, or exceeded the RNIs [38]. Pearson correlation and regression analyses were performed to assess the strength of the association between energy intake and BMI of rural and urban mothers.

## 3. Results

Rural mothers were younger than urban mothers: 35.8 (11.6) vs.44.4 (7.6) years (*p* = 0.002). However, they did not differ in BMI (28.2 (5.5) for rural and 28.2 (5.1) kg/m^2^ for urban, *p* = 0.871). Among rural mothers, 30.8% (n = 8) were normal weight, 34.6% (n = 9) overweight, and 34.6% (n = 9) obese, while, in urban mothers, 32.1% (n = 9) were normal weight, 32.1% (n = 9) overweight, and 35.7% (n = 10) obese.

The mean (SD) percentage of protein, fat, and carbohydrate contributions to total daily energy in rural versus urban mothers were as follows: protein 13.1 (2.1) vs. 13.7 (2.8)% (*p* = 0.169); fat 21.9 (6.1) vs. 25.3 (6.7)% (*p* = 0.029); and carbohydrate 62.4 (6.9) vs. 59.2 (8.3)% (*p* = 0.063). These were all within the recommended ranges (protein = 10.0 to 15.0%, fat = 15.0 to 30.0%, and carbohydrate = 55.0 to 75.0%) [22].

Energy intake was positively associated with BMI in rural and urban mothers (Figure 1). Pearson’s correlation coefficients indicated a stronger relationship in rural mothers (r = 0.77, 95% CI: 0.55, 0.89) compared to urban mothers (r = 0.58, 95% CI: 0.26, 0.78). Linear regression analysis further confirmed this association, indicating that, for every 1 kcal higher in energy intake, BMI was higher by 0.009 kg/m^2^ in rural mothers (β = 0.009, SE = 0.002, R^2^ = 0.593, *p* < 0.001) and by 0.008 kg/m^2^ in urban mothers (β = 0.008, SE = 0.002, R^2^ = 0.334, *p* = 0.001).

Table 1 presents rural and urban mothers’ mean daily energy and nutrient intakes. Overall, the mean intakes of rural mothers were higher than urban mothers, with no significant difference observed for most nutrients, apart from energy (*p* = 0.002), protein (*p* = 0.040), carbohydrate (*p* < 0.001), and iron (*p* = 0.004).

Mean daily intakes exceeded the RNIs for all nutrients examined, except for mean energy in urban mothers and mean folate in both rural and urban mothers, which were below the RNIs (Figure 2).

While all rural mothers met or exceeded the RNI for iron, a few urban mothers fell below the RNI (Table 2). About half of rural and urban mothers did not meet the RNI for folate from their diet. A few rural and urban mothers did not meet the RNIs for zinc and fiber. All rural and urban mothers met or exceeded the RNIs for vitamin A.

## 4. Discussion

We compared the energy and nutrient intakes of public health concern by rural and urban mothers to WHO’s recommended nutrient intakes using the gold-standard dietary assessment method of rWFRs. Our most important findings were that rural mothers, on average, consumed more energy than their urban counterparts, and energy intakes in rural and urban mothers were positively associated with BMI. The percentage contributions of protein, fat, and carbohydrate to daily energy intake in rural and urban mothers were within the recommended ranges. While most mothers met the RNIs for most nutrients, some did not, and disparities were observed between rural and urban mothers. Specifically, about half of rural and urban mothers did not meet the RNI for folate, indicating that dietary sources alone may be insufficient to meet their daily folate needs.

In Ghana, studies assessing dietary intake in populations have primarily used the 24 h dietary recall [15,16] and food frequency questionnaire methods [17,18]. Although these self-report methods are simple, less expensive, and more convenient to use, they are prone to misreporting [39,40,41,42]. While, in one study, the 24 h dietary recall significantly underestimated energy and nutrient intakes compared to weighed food records [19], the food frequency questionnaire overestimated intakes compared to the weighed food records in another study [21]. We observed a positive association between energy intake and BMI in both rural and urban mothers, consistent with the findings of a previous study [43]. In contrast, underreported energy intake from the 24 h dietary recall showed a weaker association with BMI [16]. Another study observed a spurious association when misreporters were included in the analysis of energy intake and its association with BMI [44]. However, after excluding the misreporters from the analysis, a positive association with BMI emerged, suggesting that misreporting energy intake can distort the relationship between energy intake and BMI. While the rWFR method used in our study provided a more accurate estimation of energy and nutrient intake, it is not feasible for assessing dietary intake in large-scale studies because it is expensive, labor-intensive, time consuming, and intrusive [39]. This underscores the need for other objective, less intrusive, and burdensome dietary assessment methods, including passive technologies for dietary intake assessment in the Ghanaian population, as they have the potential to objectively assess dietary intake in free-living individuals with less respondent burden and improved accuracy [27].

Our finding that about one-third of rural and urban mothers had a BMI in the obese category is consistent with a study that estimated the prevalence of obesity among Ghanaian women to be 35.5% [9]. As reviewed previously, the prevalence of obesity has been shown to be higher in women than men, higher in urban areas than rural areas, and rapidly increasing in rural areas compared to previous years [45]. Surprisingly, we observed that rural mothers consumed more energy than their urban counterparts. This difference likely stems from disparities in physical activity levels and occupational demands between rural and urban settings in Ghana. Rural mothers often engage in labor-intensive tasks such as farming, carrying heavy loads, and walking long distances, all of which significantly increase their daily energy requirements [46,47]. In contrast, urban mothers often work in less physically demanding roles, such as trading or office work, and tend to lead more sedentary lifestyles, relying heavily on transportation and consequently exhibiting lower total energy expenditure [46,47]. Additionally, rural mothers are often more involved in household chores than urban mothers, resulting in higher energy expenditure [46,47]. However, given the limitations of the small sample size, which may affect the generalizability of the results, this finding should be interpreted cautiously. Accordingly, the results should be considered as exploratory, highlighting the need for further studies with larger sample sizes to validate them. While energy intake in excess of expenditure is a significant contributor to the obesity pandemic, urbanization and nutrition transition have been reported as key drivers of the increasing prevalence of obesity among women in Ghana [48]. Consequently, obesity has been linked to chronic diseases, such as gestational diabetes, cardiovascular diseases, and hypertension in Ghanaian women [49,50,51]. The government of Ghana must prioritize strategies to curb urbanization and promote physical activity through healthy community planning to address the growing burden of obesity and associated chronic diseases among Ghanaian women.

Traditional Ghanaian foods are typically prepared from starchy staple foods (main sources of calories), accompanied by any protein of choice (beef, poultry, fish, eggs, etc.), and consumed with different types of soups and stews (prepared from vegetables) [52,53]. Most of these traditional Ghanaian foods are excellent sources of fiber and vitamin A because of the minimal use of processed food ingredients in their preparation [52,53]. Our finding that all rural and urban mothers met the recommended intake for vitamin A, with few rural and urban mothers not meeting the recommended fiber intake, suggests that traditional Ghanaian foods are indeed rich sources of vitamin A and fiber. Adequate consumption of fiber has several health benefits, such as reducing the risk of obesity, improving glycemic control and insulin sensitivity, and reducing the risk of type 2 diabetes and certain cancers [54]. Promoting adequate fiber intake could potentially play an essential role in preventing chronic diseases and enhancing the overall health and well-being of Ghanaian women.

All rural mothers met the recommended iron intake, but some urban mothers did not. This could partly be due to the diversity in dietary patterns in rural versus urban areas. In Ghana, rural residents usually grow iron-rich leafy green vegetables, such as *kontomire* (*Taro leaves*), typically harvested fresh for cooking, and raise livestock (e.g., cattle, goats, poultry, and pork) for household consumption [55,56]. This ensures a more consistent supply of fresh iron-rich foods that contribute to higher iron intake in rural areas than in urban areas, where residents may rely on more processed and fast foods, which are usually lower in essential micronutrients like iron [25,57]. Zinc intake was adequate in most rural and urban mothers; however, a higher prevalence of zinc and iron deficiencies has previously been reported among some Ghanaian women [10,11,58]. Therefore, dietary practices that promote adequate intake of zinc and iron, such as promoting regular consumption of traditional or fortified foods rich in these nutrients and the use of micronutrient supplements, must be encouraged in Ghanaian women to reduce the risk of deficiencies.

We found that about half of both rural and urban mothers did not meet the recommended folate intake. This indicates that foods consumed by mothers may have low folate content, consistent with a study that observed that some Ghanaian traditional foods contributed poorly to the daily folate needs of women [59]. A high prevalence of folate deficiency has been reported in Ghanaian women [13], and low birth weights, premature births, and neural tube defects have been documented as associated health consequences [60,61]. Higher prevalence of neural tube defects, such as myelomeningocele, meningocele, and cranium bifida, has been reported in some rural and urban areas in Ghana [62,63]. This calls for interventions that encourage, particularly, women of childbearing age to consume folic acid-fortified foods and supplements to minimize the risk associated with inadequate folate intake. Although some interventions have been implemented by the government of Ghana to prevent micronutrient deficiencies in the Ghanaian population, our findings indicate the need to prioritize strategies aimed at intensifying folic acid supplementation in women in rural and urban areas of Ghana because it appears diet alone may not be sufficient to meet their daily recommended folate intake.

The government of Ghana, in collaboration with civil society organizations including the WHO, UNICEF, FAO, and the Global Alliance for Improved Nutrition, has implemented some nutrition interventions to reduce micronutrient deficiencies, particularly among women and children [64]. These interventions include the mandatory fortification of wheat flour with essential micronutrients such as iron, folic acid, zinc, and B vitamins [65]; universal salt iodization [66]; vitamin A fortification [67]; and targeted micronutrient supplementation programs for women [68]. However, despite these efforts, fortified foods are only intermittently available, particularly in rural areas, and micronutrient supplementation has been largely limited to specific urban settings [64]. Our findings highlight a critical need to strengthen and prioritize targeted strategies aimed specifically at improving micronutrient access and intake, particularly folate, among women. The findings suggest that relying solely on dietary sources may be insufficient to address folate and other micronutrient deficiencies in Ghanaian women. Our results emphasize the urgency of expanding education on folic acid supplementation, particularly for women of reproductive age in both rural and urban deprived areas. Policies to integrate folic acid supplementation into routine maternal health services and ensure its consistent delivery through antenatal care programs nationwide are needed. Food fortification efforts must also be expanded to wider populations and to include more Ghanaian staple foods fortified with folic acid and other key micronutrients of public health concern. Additionally, nutrition education initiatives should emphasize the importance of dietary diversity, consumption of folate-fortified foods, and adherence to micronutrient supplementation programs. Incorporating these objectives into national nutrition policies may significantly reduce the burden of folate and other micronutrient deficiencies, thereby improving maternal and child health outcomes in Ghana.

### Study Strengths and Limitations

A major strength of our study is the use of the rWFR method, considered the gold standard for dietary intake assessment. We assessed the dietary intake of mothers on two weekdays and one weekend day to capture day-to-day variability, allowing for a more accurate and comprehensive assessment of habitual dietary intake. We also examined variations in maternal dietary intake by geographical location, providing valuable insights into how rural and urban settings influence dietary behaviors and nutrient intake patterns. This study focused on critical essential nutrients of public health concern in Ghana, providing insight into the habitual intake levels of these nutrients among the participants. However, we acknowledge several limitations that should be considered when interpreting our findings. First, this study involved a relatively small sample size and was conducted in only two communities, one rural and one urban, which limits the generalizability of the findings to other rural and urban populations in Ghana. Therefore, our findings should be considered exploratory and interpreted with caution. Second, this study did not account for seasonal variations in food availability and consumption, which could influence dietary intake, particularly energy intake, between the two geographical locations and their food acquisition practices. Third, the exclusion of essential nutrients beyond those of public health concerns may limit the comprehensiveness of the nutrient intake assessment. While we have data on other micronutrients, these are not reported in this paper. Finally, we did not assess the intake of dietary supplements, as supplement use is generally uncommon among women in Ghana [69] and was not the primary focus of the main study. As such, the reported nutrient intakes reflected only those obtained through dietary sources.

## 5. Conclusions and Future Research Directions

Energy intake was generally higher in rural mothers than urban mothers. The daily energy intakes of both rural and urban mothers were positively associated with BMI. The percentages of protein, fat, and carbohydrate contributing to the daily energy intake of rural and urban mothers were all within the recommended ranges. Overall, most rural and urban mothers met or exceeded the recommended micronutrient intakes, except for folate. Our findings indicate that dietary sources alone may be inadequate to meet the micronutrient needs of some Ghanaian women, particularly for folate. Micronutrient supplementation, especially folic acid for women of childbearing age, and food fortification programs in Ghana must be expanded and made more accessible to wider populations in both rural and urban communities. Nutrition programs aimed at increasing fruit and vegetable consumption should be intensified to help reduce the risk of micronutrient deficiencies and improve the health and overall well-being of women in Ghana. To provide a complete and holistic understanding of total nutrient intakes, future dietary assessment studies, utilizing the gold standard method employed in our study, should also evaluate dietary supplement use among women. Future studies should also include larger sample sizes across diverse geographical regions and account for seasonal variations in dietary intake. Evaluating the impact of micronutrient supplementation, food fortification, and nutrition education programs on women’s nutritional status should be a priority for additional research to inform effective policies and interventions in Ghana.

## Figures and Tables

**Figure 1 nutrients-17-02567-f001:**
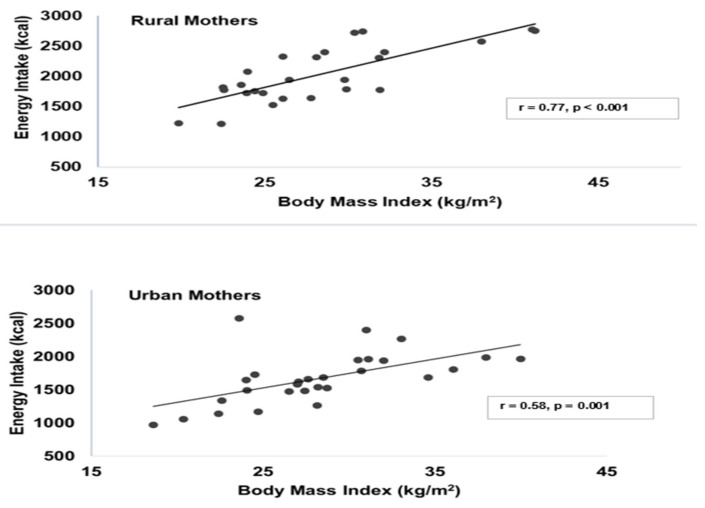
Associations between energy intake and body mass index of rural and urban mothers.

**Figure 2 nutrients-17-02567-f002:**
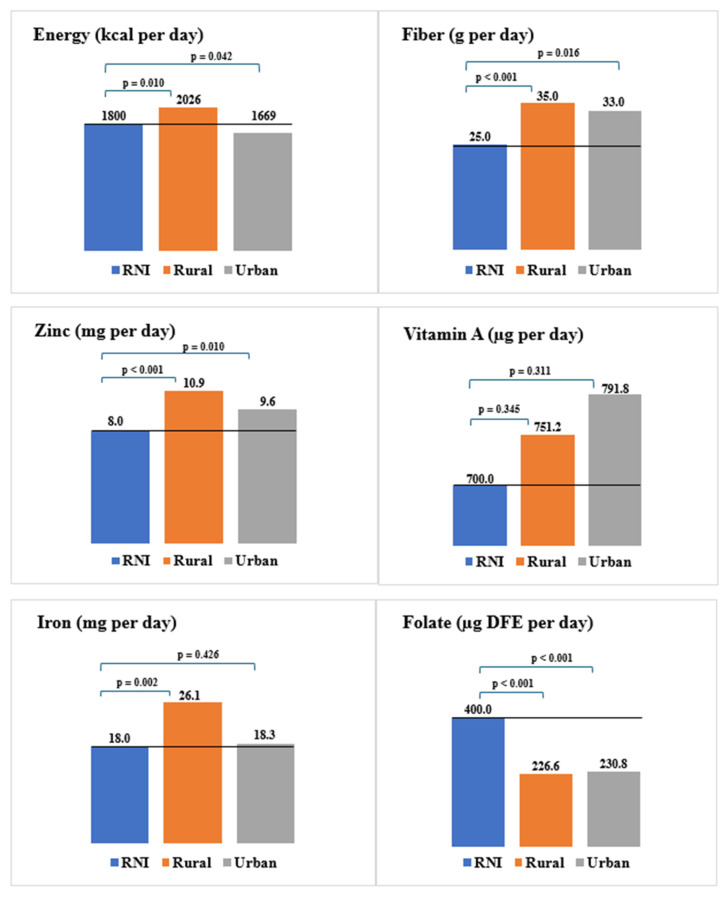
Comparison of rural and urban mothers’ mean daily energy and nutrient intakes of public health concern to recommended nutrient intakes (RNIs) (represented by a blue bar and black horizontal line). *p*-values are statistically significant at *p* < 0.05.

**Table 1 nutrients-17-02567-t001:** Comparison of mean daily energy and nutrient intakes of public health concern by rural and urban mothers in Ghana.

Outcome Variables	Rural (n = 26)	Urban (n = 28)	*p*-Value
	Mean ± SD	Mean ± SD	
Energy (kcal)	2026 ± 461	1669 ± 385	0.002
Carbohydrate (g)	325.8 ± 43.6	249.6 ± 42.2	<0.001
Protein (g)	66.6 ± 21.4	57.5 ± 12.3	0.040
Fat (g)	51.0 ± 10.6	49.1 ± 9.7	0.431
Fiber (g)	35.0 ± 11.5	33.0 ± 18.7	0.317
Zinc (mg)	10.9 ± 3.6	9.6 ± 3.4	0.089
Iron (mg)	26.1 ± 13.1	18.3 ± 7.4	0.004
Vitamin A (µg)	751.2 ± 126.5	791.8 ± 184.3	0.429
Folate (µg DFE)	226.6 ± 95.1	230.8 ± 122.6	0.444

**Table 2 nutrients-17-02567-t002:** Percentage of rural and urban mothers who fell below, met, or exceeded RNIs for energy and nutrient intakes of public health concern.

Outcome Variables	Below RNIs % (n)		Met RNIs % (n)		Exceeded RNIs % (n)	
	Rural	Urban	Rural	Urban	Rural	Urban
Energy (kcal)	7.7 (2)	21.4 (6)	42.3(14)	67.9 (20)	38.5 (10)	7.1 (2)
Fiber (g)	3.8 (1)	7.1 (2)	84.6 (22)	82.1 (23)	11.5 (3)	10.7 (3)
Zinc (mg)	7.7 (2)	10.7 (3)	80.8 (21)	78.6 (22)	11.5 (3)	10.7 (3)
Iron (mg)	0.0 (0)	14.3 (4)	76.9 (20)	82.1 (23)	23.1 (6)	3.6 (1)
Vitamin A (µg)	0.0 (0)	0.0 (0)	92.3 (24)	92.9 (26)	7.7 (2)	7.1 (2)
Folate (µg DFE)	46.2 (12)	53.6 (15)	53.8 (14)	46.4 (13)	0.0 (0)	0.0 (0)

RNIs = recommended nutrient intakes, n = frequency. RNIs are obtained from the FAO/WHO Human Energy/Vitamin and Mineral Requirements Handbooks. Z-scores for mean intakes below −1 are classified as below RNIs, between −1 and +1 meet RNIs, and above +1 exceed RNIs.

## Data Availability

The raw data supporting the conclusions of this article will be made available by the authors on request.

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
