# Peer review of "Energy and Nutrient Intakes of Public Health Concern by Rural and Urban Ghanaian Mothers Assessed by Weighed Food Compared to Recommended Intakes"

_nutrients, 2025, doi:10.3390/nu17152567_

Round 1
Reviewer 1 Report
Comments and Suggestions for Authors
The submitted manuscript presents a cross-sectional study comparing the energy and dietary intakes of rural and urban Ghanaian mothers and evaluating them to recommended nutrient intakes, using researcher-administered weighed food records. The use of an objective dietary assessment method is a notable strength, particularly in a context where self-report data are prone to misreporting. The findings offer useful preliminary insights into dietary intake; however, several improvements and clarifications are required before it can be considered for publication. Please find my suggestions and comments below:
1. The sample size in this study is relatively small, and the authors should clearly present the research as a pilot study. Corresponding revisions are needed in the Title, the Materials and Methods section, and consistently throughout the manuscript to reflect the preliminary nature of the study.
2. In Section 2.1 – Study Design (line 98). the authors should state explicitly whether ethical approval was obtained and whether participants provided informed consent.
3. In Section 2.2 – Data collection (line 110), the authors should clarify whether the daily dietary intake calculations accounted for the intake of fortified foods and/or the use of dietary supplements. This clarification is particularly important given the high prevalence of inadequate folate intake among women of reproductive age in Ghana.
4. The authors should add a new section entitled - 2.7. Dietary Assessment (line 199), in which the provide a more detailed description of how the dietary data were analysed to determine the nutritional composition of women’s diets. This section should include a more explanation of how z-score were calculated and interpreted using the probability approach, supported by appropriate references to relevant guidelines or standards.
5. The Discussion section should be strengthened by the authors through future elaborating on the public health implications of the findings. For example, the authors could discuss how the results might inform nutritional programmes or policies targeting women in rural and urban areas.
6. The end of the Discussion section, the authors should include a separate subsection entitled Study Limitations and Strengths. This should clearly outline the methodological limitations as well as the study’s strengths.
7. The final section – Conclusions (line 345) should be revised and retitled as Conclusions and Future Research Directions. The authors should provide a balanced summary of key findings and offer recommendation for further research in this area.
Author Response
RE: Manuscript ID: nutrients-3778356
Responses to Reviewer 1 Comments
Comment 1. The sample size in this study is relatively small, and the authors should clearly present the research as a pilot study. Corresponding revisions are needed in the Title, the Materials and Methods section, and consistently throughout the manuscript to reflect the preliminary nature of the study.
Response 1.
Thank you for this thoughtful comment. We acknowledge the relatively small sample size; however, we respectfully wish to clarify that this study forms part of a larger research project, with a defined study protocol published elsewhere (see Reference 27). The sample size of the larger project referenced was limited by funding. Labeling this manuscript as a pilot study would misrepresent the broader scope and intended contribution of our research. Instead, we have revised the manuscript to clearly acknowledge the small sample size as a limitation, specifically in the newly added section 4.1 (see line 353), Study Strengths and Limitations, which was created in response to your suggestion. In this section, we have emphasized the exploratory nature of the findings and stressed the importance of interpreting them with caution (see lines 360-364). We have also revised the title to reflect the revisions made throughout the manuscript. We hope this addresses your concern while preserving the context and purpose of this study within the broader research framework.
Comment 2. In Section 2.1 – Study Design (line 98). The authors should state explicitly whether ethical approval was obtained and whether participants provided informed consent.
Response 2.
Ethical approval was obtained for the study from both the University of Georgia and the University of Ghana, and all participants provided informed consent. While this information was originally included under the Institutional Review Board Statement, it has now been incorporated into Section 2.1, Study Design, following your suggestion (see lines 106-110).
Comment 3. In Section 2.2 – Data collection (line 110), the authors should clarify whether the daily dietary intake calculations accounted for the intake of fortified foods and/or the use of dietary supplements. This clarification is particularly important given the high prevalence of inadequate folate intake among women of reproductive age in Ghana.
Response 3.
Intake of dietary supplements was not assessed as this was not the focus of the main study. This information has been added (see lines 130-131). However, the software used for nutritional analysis had extensive databases of foods and beverages commonly consumed in Ghana, including fortified foods, traditional foods, processed foods, sugar-sweetened beverages, and others, programmed into it. Therefore, daily dietary intake calculations included fortified foods when consumed by a mother. This has been added in section 2.2, Data collection (see lines 128-130).
Comment 4. The authors should add a new section entitled - 2.7. Dietary Assessment (line 199), in which they provide a more detailed description of how the dietary data were analyzed to determine the nutritional composition of women’s diets. This section should include a more explanation of how z-scores were calculated and interpreted using the probability approach, supported by appropriate references to relevant guidelines or standards.
Response 4.
Thank you for your comment regarding the need for more explanation on the nutritional analysis, as well as the calculation and interpretation of Z-scores using the probability approach. Rather than creating a new section, which we believe would largely duplicate the procedures already described, we have expanded the relevant sections to provide additional detail. Specifically, we have elaborated on the nutritional analysis and Z-score calculations in Sections 2.4 (lines 148 to 178) and 2.5 (lines 193 to 201).
Comment 5. The Discussion section should be strengthened by the authors through future elaborating on the public health implications of the findings. For example, the authors could discuss how the results might inform nutritional programs or policies targeting women in rural and urban areas.
Response 5.
The Discussion (Section 4.0) has been strengthened, and we have elaborated on the public health implications of the findings in the concluding paragraph (see lines 332-352).
Comment 6. At the end of the Discussion section, the authors should include a separate subsection entitled Study Limitations and Strengths. This should clearly outline the methodological limitations as well as the study’s strengths.
Response 6.
A new section 4.1 (see line 353), Study strengths and limitations, has been added, outlining the methodological limitations as well as the study’s strengths (see lines 354-372).
Comment 7. The final section – Conclusions (line 345) should be revised and retitled as Conclusions and Future Research Directions. The authors should provide a balanced summary of key findings and offer recommendations for further research in this area.
Response 7
The final section has been revised and retitled as 5.0 Conclusions and Future Research Directions (see line 373). We have also included a balanced summary of the key findings and added recommendations for further research in the area of dietary assessment in this section (see lines 374-390).
Reviewer 2 Report
Comments and Suggestions for Authors
This manuscript presents a relevant and well-executed study comparing energy and nutrient intakes of rural and urban mothers in Ghana using researcher-conducted weighed food records (rWFR). The methodology is strong, the data are clearly presented, and the discussion is meaningful. The findings add important knowledge regarding nutritional disparities in LMICs and support future policy and intervention design. However, several changes should be reviewed:
-
Expand the rationale behind this hypothesis in the Introduction, and discuss more thoroughly why the rural intake was higher (e.g., physical activity levels, agricultural work, food access).
-
Add a stronger statement in the Discussion acknowledging this limitation and clarifying that the results should be interpreted as exploratory.
- Briefly justify the exclusion of other relevant nutrients such as calcium and vitamin D, or note this as a limitation.
Author Response
RE: Manuscript ID: nutrients-3778356
Responses to Reviewer 2 Comments
Comment 1. Expand the rationale behind this hypothesis in the Introduction, and discuss more thoroughly why the rural intake was higher (e.g., physical activity levels, agricultural work, food access).
Response 1
The rationale underlying our hypothesis has been expanded in the Introduction (see lines 84-92). Additionally, the finding of higher energy intake among rural mothers compared to urban mothers has been thoroughly discussed in section 4.0 (see lines 278-289).
Comment 2. Add a stronger statement in the Discussion acknowledging this limitation and clarifying that the results should be interpreted as exploratory.
Response 2.
We have added a stronger statement of this limitation and emphasized that the results should be interpreted as exploratory in the Discussion (see lines 286-289).
Comment 3. Briefly justify the exclusion of other relevant nutrients such as calcium and vitamin D, or note this as a limitation.
Response 3.
Our study focused on specific nutrients of public health concern in Ghana and did not include other relevant nutrients such as calcium and vitamin D. This limitation has been acknowledged in Section 4.1, Study Strengths and Limitations (see lines 367-369).
Round 2
Reviewer 1 Report
Comments and Suggestions for Authors
The authors have addressed the issues raised during the first round of peer review and have the necessary revisions in line with the reviewer comments. I have one minor suggestions:
- Reference (lines 638-660 ): during the final formatting of the manuscript, kindly ensure that reference number 62 is correctly formatted.